# Fabrication and Characterization of Fiber-Reinforced Composite Sandwich Structures Obtained by Fused Filament Fabrication Process

**George Razvan Buican** [1], **Sebastian-Marian Zaharia** [1,*], **Mihai Alin Pop** [2], **Lucia-Antoneta Chicos** [1], **Camil Lancea** [1], **Valentin-Marian Stamate** [1] and **Ionut Stelian Pascariu** [1]

[1] Department of Manufacturing Engineering, Transilvania University of Brasov, 500036 Brasov, Romania; buican.george@unitbv.ro (G.R.B.); l.chicos@unitbv.ro (L.-A.C.); camil@unitbv.ro (C.L.); stamate_vali@yahoo.com (V.-M.S.); ionut.pascariu@student.unitbv.ro (I.S.P.)

[2] Department of Materials Science, Transilvania University of Brasov, 500036 Brasov, Romania; mihai.pop@unitbv.ro

[*] Correspondence: zaharia_sebastian@unitbv.ro

**Abstract:** The application of fused filament fabrication processes is rapidly expanding in many domains such as aerospace, automotive, medical, and energy, mainly due to the flexibility of manufacturing structures with complex geometries in a short time. To improve the mechanical properties of lightweight sandwich structures, the polymer matrix can be strengthened with different materials, such as carbon fibers and glass fibers. In this study, fiber-reinforced composite sandwich structures were fabricated by FFF process and their mechanical properties were characterized. In order to conduct the mechanical tests for three-point bending, tensile strength, and impact behavior, two types of skins were produced from chopped carbon-fiber-reinforced skin using a core reinforced with chopped glass fiber at three infill densities of 100%, 60%, and 20%. Using microscopic analysis, the behavior of the breaking surfaces and the most common defects on fiber-reinforced composite sandwich structures were analyzed. The results of the mechanical tests indicated a significant influence of the filling density in the case of the three-point bending and impact tests. In contrast, the filling density does not decisively influence the structural performance of tensile tests of the fiber-reinforced composite sandwich structures. Composite sandwich structures, manufactured by fused filament fabrication process, were analyzed in terms of strength-to-mass ratio. Finite element analysis of the composite sandwich structures was performed to analyze the bending and tensile behavior.

**Keywords:** fused filament fabrication; fiber-reinforced composite sandwich structures; infill density; mechanical proprieties; finite element analysis

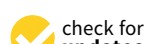



## 1. Introduction

Manufacturing parts using the process of fused filament fabrication (FFF) is considered to be one of the fastest manufacturing processes, which involves low costs, and is used for applications in industries such as aviation [1–3], automotive [4], medical [5–7], and food [8–10]. The material extrusion 3D printing process of polymer filaments (polylactic acid (PLA), polyethylene terephthalate (PET), acrylonitrile butadiene styrene (ABS), high impact polystyrene (HIPS), nylon, polyoxymethylene (POM), polycarbonate (PC), polyether ether ketone (PEEK)) has attracted a major interest from specialists in additive manufacturing techniques.

Currently, interest has shifted to combined filaments of at least two materials that have different physical and chemical properties, with properties superior to the properties of each component, thus developing the concept of composite filaments. Composite filaments are available in two versions: short fibers (discontinuous) and continuous fibers. Composite filaments can be manufactured using different types of fiber material, such as:

carbon [11–13], glass [14–17], Kevlar [18,19]. Recent studies have investigated the mechanical performance of composites manufactured by the FFF process with short or continuous carbon fibers with various polymeric matrices made from PLA [20,21], ABS [22,23], and PEEK [24].

In the manufacture of fiber-reinforced thermoplastic composites by the FFF process, certain parameters can be varied (temperature, infill density, speed, nozzle height, filament over-lap, layer thickness, infill pattern, raster angle, and fiber orientation), which can significantly influence the mechanical proprieties of printed parts [25,26]. The infill density played a minor role in the case of tensile stress tests performed on reinforced thermoplastic composites manufactured by the FFF process [27]. In contrast, in the compression tests of fiber-reinforced thermoplastic composites, the infill density significantly affected the properties of parts manufactured by the fused filament fabrication process [28].

Composite sandwich structures are a configuration of materials manufactured by gluing two thin skins on a thick and light core. The manufacturing process of sandwich structures by material extrusion 3D printing is attractive for different industries such as aerospace, marine, automotive, windmills, and civil constructions [28]. Recent research has investigated the sandwich structures manufactured by material extrusion 3D printing process, using various types of materials: PLA [29,30], PLA/wood [31], PLA/PHA [32], hemp/PLA 3D [33]. For these sandwich structures, manufactured by the FFF process, the mechanical performances were determined by various types of tests (compression, impact, bending, microhardness), and the failure modes were analyzed using microscopic analysis. Recently, research has been conducted on the feasibility of manufacturing composite sandwich structures made from carbon-fiber-reinforced thermoplastic using the material extrusion 3D printing process. The results of the manufactured 3D sandwich structures showed that 3D CFRTP (carbon-fiber-reinforced thermoplastic) sandwich structures can be 3D printed using different core shapes and a higher roughness than of PLA. Moreover, three-point bending tests of sandwich structures manufactured with different core shapes have shown that their mechanical properties depend largely on the shape of the core [34]. Extensive research has been conducted to investigate smart continuous carbon fiber thermoplastic lattice truss sandwich structure manufactured by the FFF process. The conclusion was that truss inclination angle had a significant influence on the mechanical properties of sandwich structures [35]. A new sandwich structure has been proposed [36] using additive manufacturing processes (3D printing using material extrusion) from ABS laminated with CFRP layers. The purpose of developing such a sandwich structure was to improve the specific strength and stiffness for parts used for aviation (clamps in the structure of a quadcopter) industry.

Many research works [12,20,25,37,38] have been performed on carbon fiber analysis in order to identify the main defects from 3D printed parts through the FFF process. The identified defects of composite filaments that appeared during the FFF process were inter-raster gaps in the structure, uneven distribution of fibers, poor bonding of layers, and breaking of fibers. These defects that appear on printed parts made from fiber-reinforced composite can be summarized as distortion of the shape generated by residual stresses from uneven temperature gradients, uneven distribution of fiber material in thermoplastic filaments reinforced with composite fiber, gaps found in the matrix or filament, poor adhesion between fiber and matrix. In a recent study [39], these defects were identified in 3D printed parts made from continuous composite fibers through scanning electron microscopy. The most pronounced defects of the 3D printed parts were the upper surface roughness and porosity and the poor connection between the fiber and the matrix.

At present, 4D printing technology has been implemented in order to explore meta-sandwich structures with reversible energy absorption. This concept was based on the positioning of soft hyperelastic polymers with shape memory to design auxetic structures from two materials. The models, analyzed using finite element method, were developed to accurately simulate the behaviors of 4D printed meta-structures in a loading–unloading compression test cycle [40].

Following the analysis of the current state of research in the relevant fields, it can be noted that there are scientific challenges of an interdisciplinary nature (additive manufacturing and aviation) which determine the implementation of composite sandwich structures manufactured by material extrusion 3D printing process. In this paper, light sandwich structures were manufactured flatwise (in the YX plane) with two types of carbon-fiber-reinforced polymer (CFRP) skins and glass-fiber-reinforced polymer (GFRP) core using the material extrusion 3D printing process at three infill densities of 100%, 60% and 20%. The mechanical performances of composite sandwich structures were determined using mechanical tests (tensile, three-point bending and impact), and the failure modes of the composite sandwich specimens, manufactured by the FFF process, were determined by microscopic analysis. A comparative study was performed between the values of the reaction forces and the failure mode that appeared in the experimental tests (three-point bending and tensile) and the values of the reaction forces that appeared in the models analyzed with the numerical methods. The purpose of manufacturing, through the FFF process, of composite sandwich structures from carbon fiber and glass fiber filaments was to implement such structures on the surface of a UAV model, where the loads are higher (wing leading edge, electric engine nacelle).

## 2. Materials and Methods

### 2.1. Materials Properties and Manufacturing Conditions of the Sandwich Specimens

Within this study, the skins of the sandwich structures were manufacture using two types of carbon-fiber-reinforced polymer (CFRP). For the manufacture of sandwich structures cores, through the FFF process, only one type of filament was used namely glass-fiber-reinforced polymer (GFRP). From this point forward the two sandwich specimens will have the following abbreviations: CFRP1-GFRP and CFRP2-GFRP. The CFRP1-GFRP sandwich specimen has two skins made from filaments with carbon fiber and the core made from filaments with fiberglass. The mechanical and thermal properties, as supplied by the manufacturer, for the carbon fiber filament Novamid ID1030 CF10 (DSM GmbH, Braunschweig, Germany) are shown in Table 1. This filament is a nylon PA6/66 filament but mixed with 10% chopped carbon fiber. The mechanical properties of the carbon fiber filament ColorFabb PA-CF (Belfeld, The Netherlands) used in the second type of sandwich specimen (CFRP2-GFRP) in Table 2 are described. The ColorFabb PA-CF filament consists of 10% chopped carbon fiber strands in a polyamide matrix. The mechanical properties of the materials were provided by the filament manufacturers. All three types of composite filament were dried according to the manufacturers' specifications.

**Table 1.** Mechanical properties of the filament used in 3D printing of sandwich specimen skins (CFRP1).

| Mechanical Properties | NOVAMID ID 1030 CF10 | Standard |
|---|---|---|
| Tensile Strength [MPa] | 110 | ISO 527 [41] |
| Tensile modulus [MPa] | 7630 | ISO 527 |
| Strain at yield [%] | 2.5 | ISO 527 |
| Density [g/cm$^3$] | 1.17 | ISO 1183 [42] |

**Table 2.** Mechanical properties of the filament used in 3D printing of sandwich specimen skins (CFRP2).

| Mechanical Properties | ColorFabb PA-CF | Standard |
|---|---|---|
| Tensile Strength [MPa] | 107 | ISO 527 |
| Tensile modulus [MPa] | 8110 | ISO 527 |
| Strain at yield [%] | 2 | ISO 527 |
| Density [g/cm$^3$] | 1.4 | ISO 1183 |

PLA glass-reinforced filament (Filamania Kft., Miskolc, Hungary) was used for the manufacture of the sandwich structures cores using the FFF process. The mechanical properties (provided by manufacturer) of the fiberglass filament can be found in Table 3.

**Table 3.** Mechanical properties of the filament used in 3D printing of sandwich specimen cores (GFRP).

| Mechanical Properties | PLA Glass-Reinforced | Standard |
|---|---|---|
| Tensile strength [MPa] | 57 | ASTM D638 [43] |
| Tensile modulus [MPa] | 4000 | ASTM D638 |
| Strain at yield [%] [%] | 3.4 | ASTM D638 |

Composite sandwich specimens were manufactured using Creat Bot DX-3D double nozzle printer (Henan Suwei Electronic Technology Co., Zhengzhou, China). Creat Bot DX-3D is a 3D printer with dual extrusion and a build volume of 250 mm × 250 mm × 300 mm. The parameters used in the material extrusion 3D printing process of composite sandwich specimens in Table 4 are presented. For the three types of tests carried out in this study (bending, tensile and impact), infill density was the main manufacturing parameter analyzed. Therefore, for the sandwich specimens manufactured by the FFF process, the infill densities used were the following 100%; 60%, and 20%.

**Table 4.** FFF manufacturing parameters for the composite sandwich specimens.

| Parameter | CFRP1 | CFRP2 | GFRP |
|---|---|---|---|
| Layer height [mm] | 0.2 | 0.2 | 0.2 |
| Print speed [mm/s] | 40 | 40 | 40 |
| Extrusion Temperature [°C] | 250 | 260 | 230 |
| Building plate temperature [°C] | 75 | 75 | 75 |
| Filament diameter [mm] | 2.85 | 2.85 | 2.85 |

The positioning on the printing table (flat XY) of all composite sandwich specimens, manufactured by the FFF process, can be seen in Figure 1.

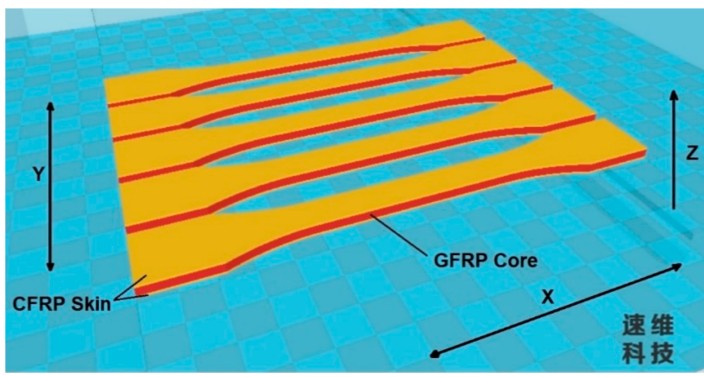

**Figure 1.** Positioning of composite sandwich specimens (XY flat) used in tensile tests.

## 2.2. Mechanical Testing

Three-point bending, tensile, and impact testing of sandwich structure specimens were performed on the WDW-150S test machine (Jinan Testing Equipment IE Corporation, Jinan, China). For testing of CFRP1-GFRP and CFRP2-GFRP composite sandwich structures, five specimens were made for each type of test (three-point bending, tensile and impact) with three different infill densities (100%, 60%, and 20%). Thus, 30 specimens (15 for CFRP1-GFRP sandwich specimens and 15 for CFRP2-GFRP sandwich specimens) were manufactured for each category of tests, with the three infill densities (100%, 60%, and 20%),

a total of 90 specimens. For the three-point static bending tests, 30 sandwich specimens were tested, 15 from each category (CFRP1-GFRP and CFRP2-GFRP), according to ASTM C393 [44], at a crosshead speed of 5 mm/min. For the two categories of specimens the infill density was varied to 100%, 60%, and 20%, and square grid was the infill pattern used. The composite sandwich specimens were placed on the supporting pins (radius R = 15 mm) with a distance of 110 mm between each other. The main dimensional characteristics of the sandwich specimens (CFRP1-GFRP and CFRP2-GFRP) tested in a static regime, at three-point bending, are presented in Table 5.

**Table 5.** Dimensions of composite sandwich specimens tested in three-point static regime.

| Material | Length L (mm) | Thickness d (mm) | Width b (mm) | Span Length S (mm) | Core Thickness c (mm) | Skin Thickness t (mm) |
|---|---|---|---|---|---|---|
| CFRP1-GFRP | 160 | 4 | 16 | 110 | 2 | 1 |
| CFRP2-GFRP | 160 | 4 | 16 | 110 | 2 | 1 |

The virtual model used in the three-point bending test of composite sandwich specimens is described in Figure 2a. Figure 2b shows a cross-sectional view of the tested composite sandwich structure. Figure 2c shows the three-point bending test setup.

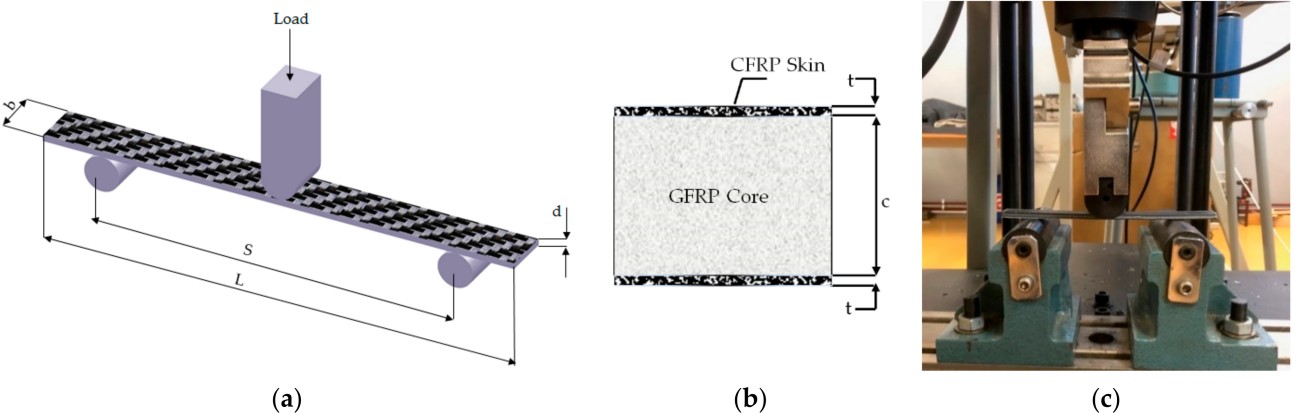

(**a**) (**b**) (**c**)

**Figure 2.** Mechanical tests: (**a**) virtual model; (**b**) cross section of the composite sandwich specimens; and (**c**) three-point bending test setup.

For the tensile tests, 30 sandwich specimens were tested, 15 from each category (CFRP1-GFRP and CFRP2-GFRP) according to the ASTM D638 standard and at a crosshead speed of 5 mm/min. In the two categories of specimens, the infill density was varied (100%, 60%, and 20%). The physical model used in the tensile strength test of composite sandwich specimens is described in Figure 3a. Figure 3b shows a cross-sectional view of the composite sandwich structure used in the tensile strength test. The setup for the tensile strength test of the composite sandwich specimens in Figure 3c is illustrated. The dimensional characteristics of the sandwich specimens (CFRP1-GFRP and CFRP2-GFRP), tensile strength tested, in Table 6 are exemplified.

For shock impact testing with a steel ball, we used the device presented in Figure 4, designed and built in our research center. The device consists of a support holding for a steel ball, a high-resolution camera, standard line, and a rigid steel support to avoid vibratory motion. When the ball is released from its support, it falls from a height of 1 m and impacts the sample. The high-resolution camera is used to measure the recoil height and the recoil energy of the sample. Details of the impact shock energy determination and calculation can be found in the paper [45]. Characteristics of the testing device: steel ball with m = 99.81 g; rigid steel support with m1 = 3.5 kg.; standard line; high resolution

camera; device to release the steel ball. The specimens had the following dimensions: length 50 mm, width 40 mm, thickness 4 mm.

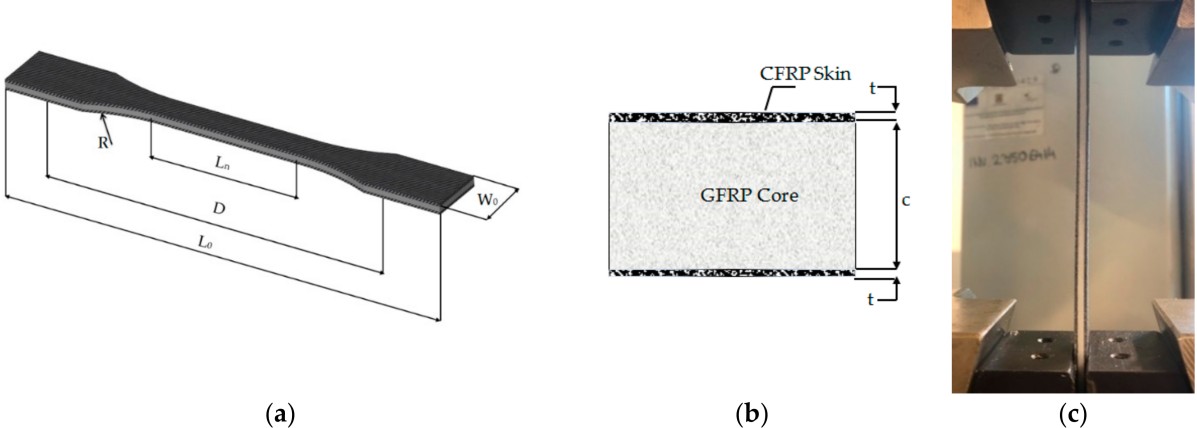

|  |  |  |
|---|---|---|
| (**a**) | (**b**) | (**c**) |

**Figure 3.** Mechanical tests: (**a**) dimensions of composite sandwich specimens; (**b**) cross section of the composite sandwich specimens; and (**c**) tensile strength test setup.

**Table 6.** Dimensions of composite sandwich specimens tested in tensile strength.

| Material | Length Overall $L_0$ (mm) | Distance between Grips D (mm) | Length of Narrow Section $L_n$ (mm) | Radius of Fillet R (mm) | Width $W_0$ (mm) | Core Thickness c (mm) | Skin Thickness t (mm) |
|---|---|---|---|---|---|---|---|
| CFRP1-GFRP | 165 | 115 | 57 | 76 | 19 | 2 | 1 |
| CFRP2-GFRP | 165 | 115 | 57 | 76 | 19 | 2 | 1 |

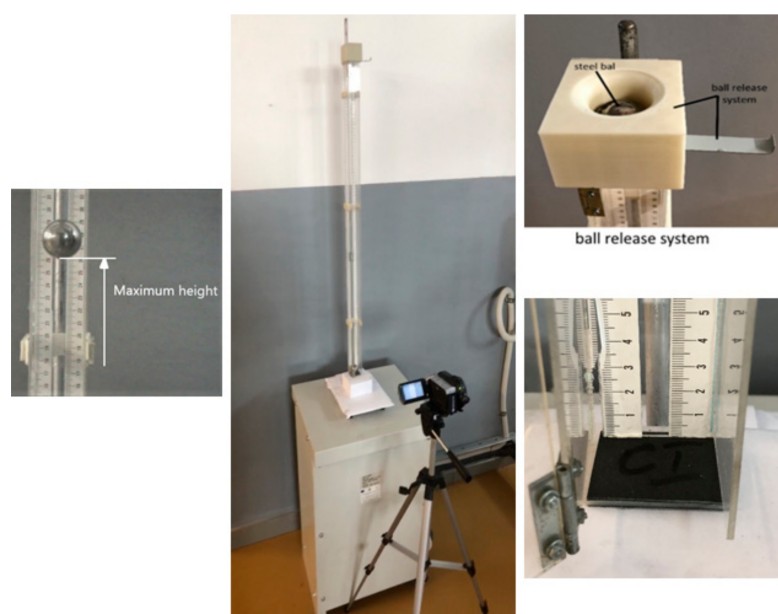

**Figure 4.** Shock impact device: impact testing experimental setup.

To perform the impact analysis of the specimens (Figure 5a), 30 unnotched specimens (from the two sandwich categories CFRP1-GFRP and CFRP2-GFRP), at with various infill densities (100%, 60%, and 20%), were tested using a Charpy hammer (Web Werkstoffprüf-maschinen, Leipzig, Germany). Figure 5b shows a cross-sectional view of the composite

sandwich structure tested at impact. The impact test setup of the composite sandwich specimens in Figure 5c is presented.

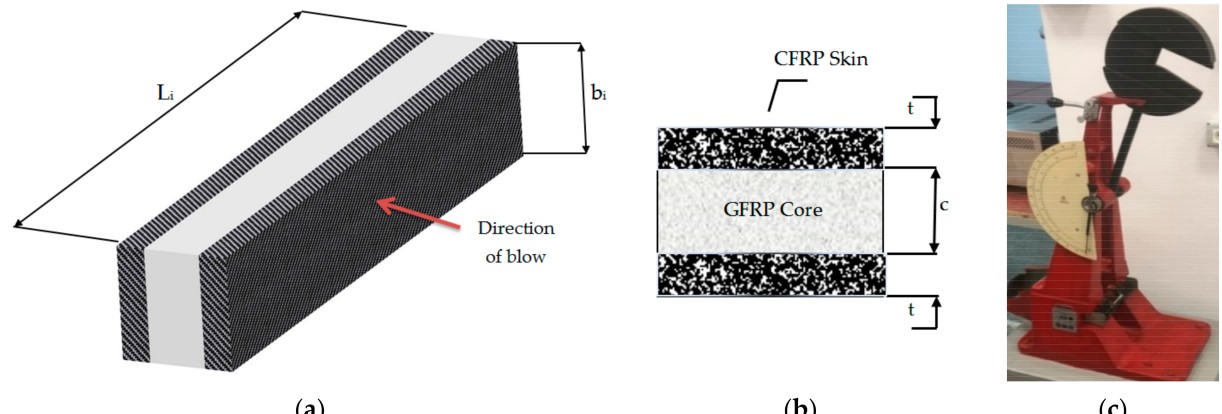

(**a**)　　　　　　　　　　　　　　　　　　(**b**)　　　　　　　　　　　　(**c**)

**Figure 5.** Mechanical tests: (**a**) dimensions of composite sandwich specimens; (**b**) cross section of the composite sandwich specimens; and (**c**) impact test setup.

For impact testing of composite sandwich specimens, a Charpy hammer was used with the following characteristics: hammer weight (66.55 N), arm length (380 mm), and initial potential energy (49 J). The impact test specimens of the composite sandwich were prepared according to ISO 179-1 standard [46] (Table 7).

**Table 7.** Dimensions of the impact composite sandwich specimens.

| Material | Length $L_i$ (mm) | Width $b_i$ (mm) | Core Thickness c (mm) | Skin Thickness t (mm) |
|---|---|---|---|---|
| CFRP1-GFRP | 55 | 10 | 5 | 2.5 |
| CFRP2-GFRP | 55 | 10 | 5 | 2.5 |

### 2.3. Statistical Analysis

Statistical analysis of data obtained from mechanical tests (tensile, three-point bending, Charpy impact, and shock impact) was performed using the main statistical indicators: mean, standard deviation, and coefficient of variation (CV). The coefficient of variation is a statistical indicator of the experimental data dispersion and represents the ratio between the standard deviation and mean. The lower the coefficient of variation, the greater the homogeneity of the data and the smaller the variation. Determining the coefficient of variation was an important aspect because the relative measurement of variation allows the comparison of different experimental data. For each type of mechanical test (tensile, three-point bending, Charpy impact, and shock impact) a total of five tests were performed, as is indicated by the test regulations in effect.

## 3. Results and Discussion

### 3.1. Three-Point Bending Behavior of the Fiber-Reinforced Composite Sandwich Structures

The results shown in Figure 6 represent the mean values of the bending strength and bending modulus, corresponding to each set of specimens, tested at bending in three points. Bending tests were performed on the two types of sandwich structures (CFRP1-GFRP and CFRP2-GFRP) at the three infill densities (100%, 60%, and 20%). Each set of specimens, tested at three-point bending, has a total of 5 composite sandwich specimens. The specific results of bending strength and bending modulus were automatically determined by the test machine software using the dimensions of the sandwich specimens. After testing the sandwich structures CFRP1-GFRP, it was found that there is a significant decrease in

bending strength of approximately 40% between the specimens with an infill density of 100% and the specimens with an infill density of 20%.

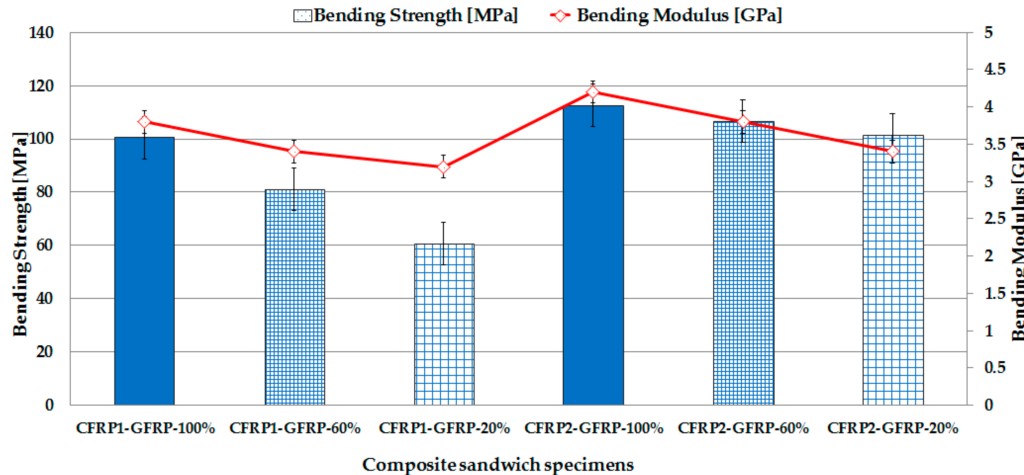

**Figure 6.** Three-point bending tests results. Results obtained for bending strength and bending modulus of the 3D printed composite sandwich.

In contrast, CFRP2-GFRP sandwich structures showed a decrease in bending strength of approximately 10% between specimens with an infill of 100% and those with an infill of 20%. Another important aspect, observed after testing the sandwich specimens, was that the bending strength of CFRP2-GFRP sandwich structures was approximately 11% higher than that of CFRP1-GFRP sandwich structures when 100% infill density was used. The bending modulus showed a lower decrease for both sandwich structures CFRP1-GFRP (approximately 16%) and CFRP2-GFRP (approximately 20%) between specimens with an infill of 100% and specimens with a 20% infill.

The value of the bending strength obtained in this paper for sandwich structures showed a higher value compared to the value of the bending strength for specimens made from a single type of material. In recent studies, the bending strength of short carbon-fiber-reinforced specimens was between 55.3 MPa [12] and 73.6 MPa [47]. In contrast, in this study, by using a composite sandwich structure consisting of carbon fiber skin and fiberglass core, the bending strength showed increase values between 100–112 MPa. It can also be seen that the bending strength of the tested sandwich specimens, built with a filling density of 20%, has a higher value compared to the bending strength values obtained in previous studies. Thus, the filling density is one of the key parameters of the FFF process as it allows the printing of sandwich structures with a density of less than 100% which, in practical terms, reduces printing time and saves material.

The most important statistical parameters (mean, standard deviation, coefficient of variation), specific to three-point bending tests, were determined for all 6 sets of the composite sandwich specimens according to the statistical methodology provided in the ASTM C393 standard. Table 8 presented the statistical indicators for each data set (bending strength and bending modulus). From the data in Table 8, it can be seen that the bending tests show homogeneous results, and the coefficient of variation shows values between 4.557% and 16.117%. A value for the coefficient of variation of up to 30% indicates homogeneity of the analyzed data. In conclusion, the mean is representative for the experimental data obtained from the three-point bending tests.

**Table 8.** Statistical parameters determined from the bending test of composite sandwich specimens.

| Sandwich Specimen | Standard Deviation (s) | Coefficient of Variation (δ)% |
|---|---|---|
| CFRP1-GFRP—100% Bending Strength [MPa]/Bending Modulus [GPa] | 6.107/0.447 | 6.082/11.763 |
| CFRP1-GFRP—60% Bending Strength [MPa]/Bending Modulus [GPa] | 3.701/0.548 | 4.557/16.117 |
| CFRP1-GFRP—20% Bending Strength [MPa]/Bending Modulus [GPa] | 5.119/0.447 | 8.419/13.968 |
| CFRP2-GFRP—100% [MPa] Bending Strength [MPa]/Bending Modulus [GPa] | 8.019/0.447 | 7.121/10.642 |
| CFRP2-GFRP—60% [MPa] Bending Strength [MPa]/Bending Modulus [GPa] | 3.130/0.447 | 2.936/11.763 |
| CFRP2-GFRP—20% [MPa] Bending Strength [MPa]/Bending Modulus [GPa] | 8.792/0.548 | 8.653/16.117 |

The bending behaviors of the two types of composite sandwich specimens (CFRP1-GFRP and CFRP2-GFRP) with different infill densities (100%, 60%, and 20%) are shown in Figure 7 in the form of load–displacement curves. In Figure 7, the load–displacement curves of all studied sandwich specimens showed an initial stage with linear behavior. At the end of this linear stage, the load–displacement curves display a nonlinear transition that determines a yielding state and then a viscoplastic response, followed by the breaking of the sandwich specimens. In Figure 7, it can be seen that the maximum force for the two types of specimens (CFRP1-GFRP and CFRP2-GFRP) is found at the infill density of 100%. The displacement showed values between 16 and 19 mm. The maximum displacement value was obtained for the CFRP2-GFRP specimen at an infill density of 20%.

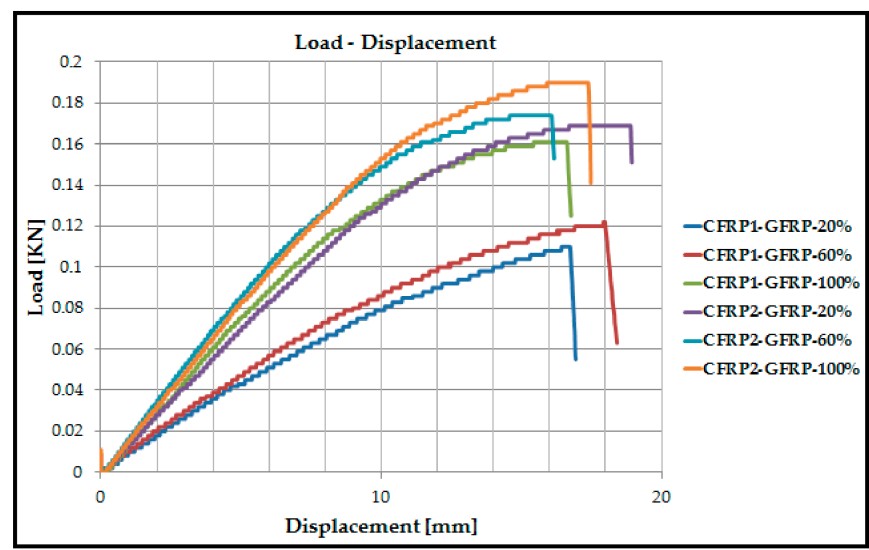

**Figure 7.** Load–displacement curves obtained from the bending test of the sandwich specimens.

*3.2. Tensile Performances of the Fiber-Reinforced Composite Sandwich Structures*

A comparative study was performed to see the effect of infill density on the tensile strength performance of the two types of composite sandwich specimens (CFRP1-GFRP and CFRP2-GFRP) manufactured by the FFF process. For this purpose, tensile strength tests were performed with an infill density of 100%, 60% and 20%, and the results (tensile strength and tensile modulus) are presented in Figure 8.

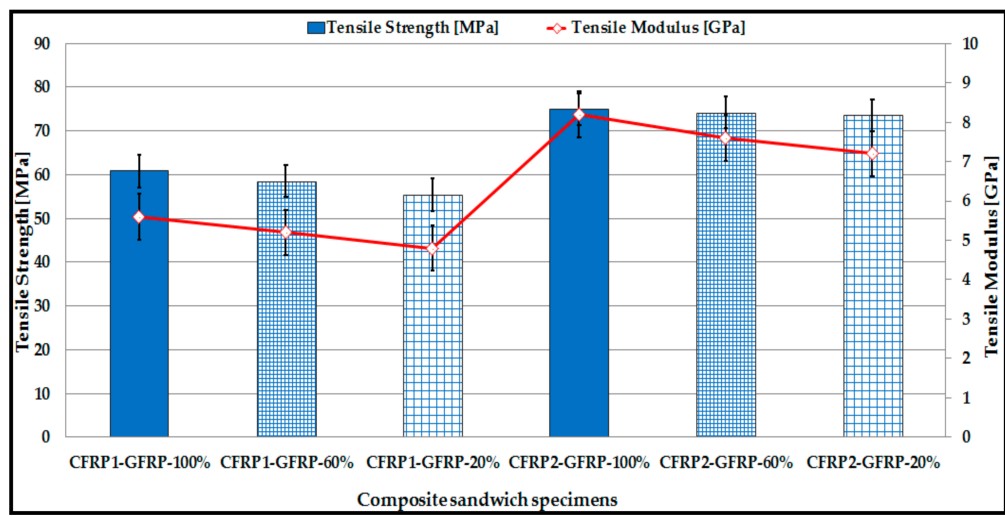

**Figure 8.** Tensile strength test results. Results obtained for tensile strength and tensile modulus of the 3D printed composite sandwich.

Composite sandwich structures with an infill density of 100% presented the best performances in the tensile strength tests. Another interesting aspect is that, in the case of both types of composite sandwich specimens (CFRP1-GFRP and CFRP2-GFRP), the values of tensile strength are very close for all the three infill densities analyzed. Therefore, it can be stated that variation in the infill density, from 100% to 20%, does not show significant differences in terms of tensile strength (for sandwich specimens CFRP1-GFRP—about 9%, and for sandwich specimens CFRP1-GFRP—about 2%).

In contrast, a lower infill density results in lower 3D printing costs and should be seen as an option, if conditions allow it, as seen in a previous study [27]. The filling density has a strong impact on the tensile performance of the sandwich specimens, both on the tensile modulus and on the tensile strength, when its value varies from 20% to 100%, as can be seen in other studies [47–49].

The low variation in the tensile strength for the sandwich specimens, observed between the three filling densities, may be due to the fact that in the tensile test, it is not the thickness of the shell that matters but rather the direction of loading. The variation in the tensile strength at the three densities is low because the specimens were loaded on the 3D printing direction.

As it can be seen in Figure 8, the mean tensile strength for the CFRP1-GFRP sandwich specimens varies between 55.4 and 60.8 MPa, and the tensile modulus is between 4.8 and 5.6 GPa. The tensile strength, measured at the same infill density (100%) for the CFRP2-GFRP sandwich specimens, was approximately 19% higher compared to that of the CFRP1-GFRP specimens.

The maximum values of the coefficient of variation (Table 9) for the two sets of experimental data (tensile strength and tensile modulus) are relatively low CV = 5.812% (for tensile strength) and CV = 9.785% (for tensile modulus) and, therefore, it can be appreciated that the experimental data obtained from the tensile strength tests is homogeneous, and the mean is representative.

The load–displacement curves obtained from the tensile strength tests of the two types of sandwich specimens (CFRP1-GFRP and CFRP2-GFRP), manufactured with infill densities of 100%, 60%, and 20%, can be seen in Figure 9. Analyzing these curves, one can appreciate a linear evolution of the sandwich specimens at tensile strength, until reaching the failure load. From Figure 9, it can be seen that the maximum force and maximum displacement, for the two types of specimens (CFRP1-GFRP and CFRP2-GFRP), is found at an infill density of 100%. This irregularity shown on the Figure 9 is due to the behavior

of the sandwich structures at the boundary between the elastic domain and the plastic domain. After this point, the curve showed a linear increase.

**Table 9.** Statistical parameters determined from the tensile strength test of composite sandwich specimens.

| Sandwich Specimen | Standard Deviation (s) | Coefficient of Variation (δ)% |
|---|---|---|
| CFRP1-GFRP—100% Tensile Strength [MPa]/Tensile Modulus [GPa] | 2.168/0.548 | 3.565/9.785 |
| CFRP1-GFRP—60% Tensile Strength [MPa]/Tensile Modulus [GPa] | 1.817/0.447 | 3.100/8.596 |
| CFRP1-GFRP—20% Tensile Strength [MPa]/Tensile Modulus [GPa] | 1.140/0.447 | 2.057/9.312 |
| CFRP2-GFRP—100% [MPa] Tensile Strength [MPa]/Tensile Modulus [GPa] | 2.915/0.447 | 3.886/5.451 |
| CFRP2-GFRP—60% [MPa] Tensile Strength [MPa]/Tensile Modulus [GPa] | 2.775/0.548 | 3.739/7.210 |
| CFRP2-GFRP—20% [MPa] Tensile Strength [MPa]/Tensile Modulus [GPa] | 4.278/0.447 | 5.812/6.208 |

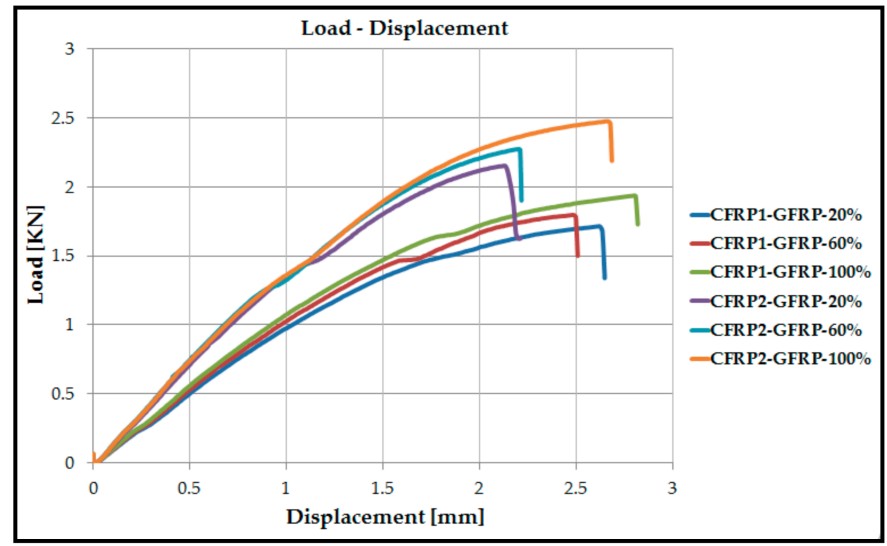

**Figure 9.** Load–displacement curves obtained from the tensile strength test of the composite sandwich specimens.

### 3.3. Mechanical Shock Properties of the Fiber-Reinforced Composite Sandwich Structures

As can be seen from Figure 10, high values for impact energy absorption were registered in the fiber-reinforced composite sandwich structures with 100% infill configurations for both materials. The impact energy increases as the porosity and the number of voids from the internal structure decrease. Another factor that has led to the improvement of these properties is that polyamide filament has 15% (CFRP2-GFRP) higher mechanical properties compared to the PA6/66 copolymer (CFRP1–GFRP). Thus, CFRP2-GFRP sandwich specimens showed higher energy absorption capacity compared to CFRP1–GFRP sandwich specimens.

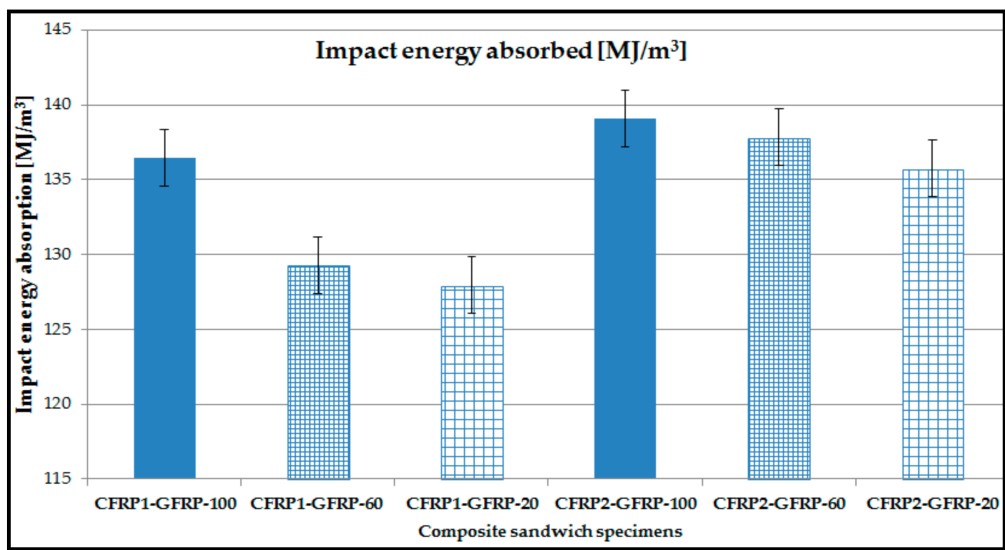

**Figure 10.** Impact shock test results. Results obtained for impact energy absorption of the 3D printed composite sandwich.

*3.4. Impact Properties of the Fiber-Reinforced Composite Sandwich Structures*

First, it is found that the impact strength of CFRP1-GFRP sandwich specimens is 13% lower than that obtained for CFRP2-GFRP sandwich specimens. As shown in Figure 11, the impact strength at different infill densities differs significantly. Thus, for both types of composite sandwich structures, the impact strength, with an infill density of 100%, is higher at about 55% compared to the specimens with an infill density of 20%. Figure 11 shows the values of impact strength for the structure CFRP1-GFRP of 18.437 kJ/m$^2$ (infill density 20%) and 33.539 kJ/m$^2$ (infill density 100%), and for the CFRP2-GFRP structure, of 21.771 kJ/m$^2$ (infill density 20%) and 38.638 kJ/m$^2$ (infill density 100%).

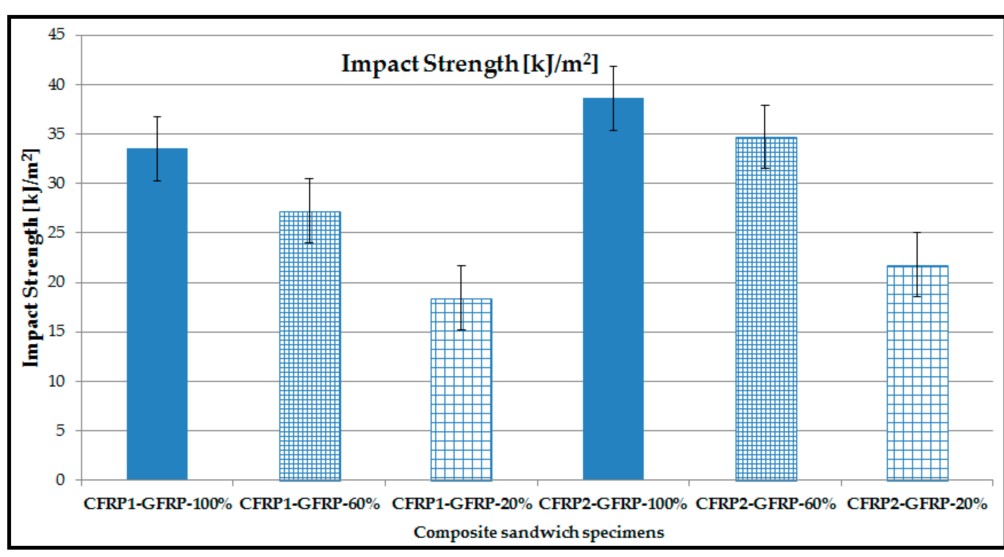

**Figure 11.** Impact test results. Results obtained for impact strength of the 3D printed composite sandwich.

The results of the experimental tests show that the infill density plays a vital role in determining the impact performances, in the case of composite sandwich structures, manufactured by the FFF process. According to the data presented in Table 10, the uncertainty of the data set (impact strength) of the composite sandwich specimens, manufactured by the

FFF process, is relatively low. The maximum value of the coefficient of variation is 20.735% and, thus, it can be estimated that the mean is representative for the experimental data set.

**Table 10.** Statistical parameters determined from the impact test of composite sandwich specimens.

| Sandwich Specimen | Standard Deviation (s) | Coefficient of Variation (δ)% |
|---|---|---|
| CFRP1-GFRP—100% Impact Strength [kJ/m²] | 5.254 | 15.665 |
| CFRP1-GFRP—60% Impact Strength [kJ/m²] | 4.353 | 15.967 |
| CFRP1-GFRP—20% Impact Strength [kJ/m²] | 3.823 | 20.735 |
| CFRP2-GFRP—100% Impact Strength [kJ/m²] | 3.643 | 9.428 |
| CFRP2-GFRP—60% Impact Strength [kJ/m²] | 3.959 | 11.403 |
| CFRP2-GFRP—20% Impact Strength [kJ/m²] | 2.442 | 11.216 |

*3.5. Strength-to-Mass Ratio of the 3D Printed Sandwich Structures*

The strength-to-mass ratio analysis (Figure 12) of the two types of sandwich structures (CFRP1-GFRP and CFRP2-GFRP) was performed for the three types of tests (three-point bending tests, tensile strength tests, and impact tests) at the three infill densities (100%, 60%, and 20%). To calculate this ratio, the mean mass of the sandwich structures and the mean strength to the three types of tests (bending, tensile, and impact) were used.

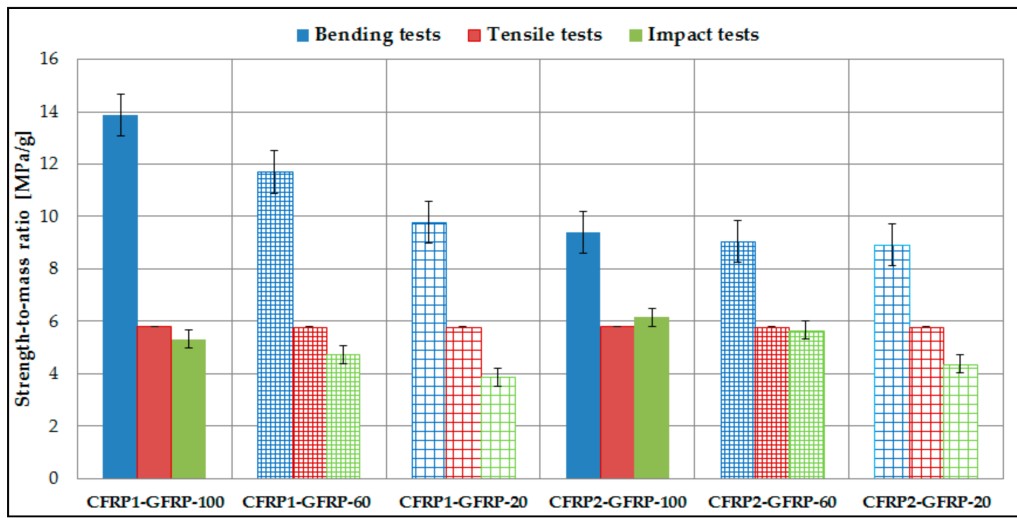

**Figure 12.** Strength-to-mass ratio of 3D printed sandwich structures.

Analyzing the composite sandwich specimens, from the point of view of the strength-to-mass ratio (Figure 12), the following conclusions can be drawn:

- In the case of three-point bending tests, for CFRP1-GFRP sandwich structures, the strength-to-mass ratio is directly proportional to the infill density. In contrast, for the second structure CFRP2-GFRP, the ratio values are very close, a fact which indicates that a lower infill density can be used, maintaining approximately the same value of the strength-to-mass ratio. For this structure, using a lower infill density reduces the time and costs of 3D printing.

- In the case of tensile strength tests, both sandwich structures have very close strength-to-mass ratio values. Thus, it can be highlighted that the infill density does not significantly change the values of the strength-to-mass ratio.
- In the case of impact tests, the results have shown that the higher the infill density, the higher the strength-to-mass ratio. Therefore, it can be stated that the value of the strength-to-mass ratio is directly proportional to the value of impact strength.

### 3.6. Microscopic Analysis of Breaking Mode and Manufacturing Defects of the Fiber-Reinforced Composite Sandwich Structures

Microscopic analysis of the failure mode for composite sandwich specimens subjected to three-point bending tests was performed with a Nikon Eclipse MA 100 metallographic microscope (Nikon Corp., Tokyo, Japan). For the structure and morphology analysis, representative samples have been cross-sectioned, embedded into acrylic resin, and leveled using an automatic metallographic Phoenix Beta polishing device from Buehler (with $Al_2O_3$ suspension and 0.05 µm grit).

In Figure 13a, the composite sandwich structure CFRP1-GFRP built with a filling density of 100%, subjected to tensile tests, was analyzed microscopically. The failure of the sandwich structure showed the following steps: debonding between the upper skins and the core, a complete cracking of the fiberglass core (Figure 13a). Cracks appeared in the lower part of the core (Figure 13b) between the extruded glass filament layers. A lesser debonding of the core from the lower skin was observed. Moreover, in Figure 13b, you can see the breaking and the arrangement of the glass fibers according to the direction of the tensile load.

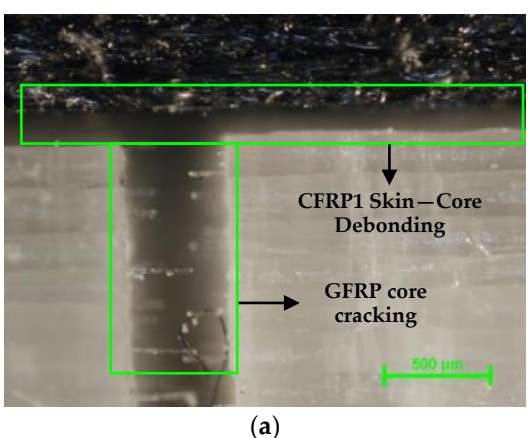
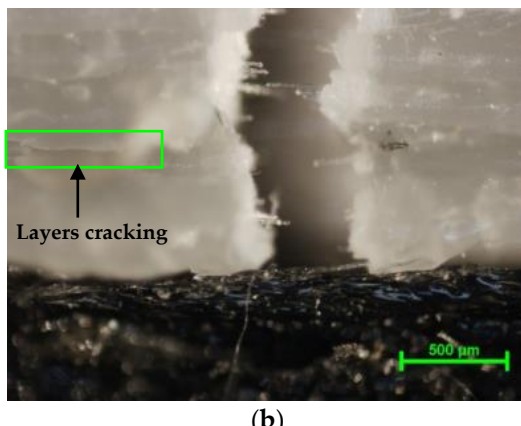

(**a**)                                                       (**b**)

**Figure 13.** Failure modes of composite sandwich structure under tensile testing (magnification 50×): (**a**) debonding between the upper skins and the core and complete cracking of the core; (**b**) cracking between the extruded glass filament layers.

In Figure 14a, the CFRP1-GFRP composite sandwich structure, subjected to three-point bending tests, built with a filling density of 100%, was analyzed microscopically. The failure of the sandwich structure, tested at three-point bending, showed debonding between the upper skins and the core. Thus, the upper carbon fiber skin was subjected to compressive load, while the lower skin was subjected to tensile load. Debonding of the upper skin from the core of sandwich structures is a critical failure mode, and once debonding occurs, the load-bearing capacity of a composite sandwich structure decreases rapidly. Next, a crack in the fiberglass core was identified (Figure 14a), and the subsequent propagation of the crack to the lower skin (Figure 14b). The propagation of the crack and the complete rupture of the core caused a debonding between the core and the lower skin. From the main mode of failure (shearing and breaking of the core) of composite sandwich structures, it can be stated that the skins are adequately sized, thick enough, and strong, they remain unaffected after performing the bending tests. Due to material defects, especially inter-track gaps, cracks propagated in the direction of load, but a shear buckling

of the core was also identified in the case of sandwich structures tested for bending at three points.

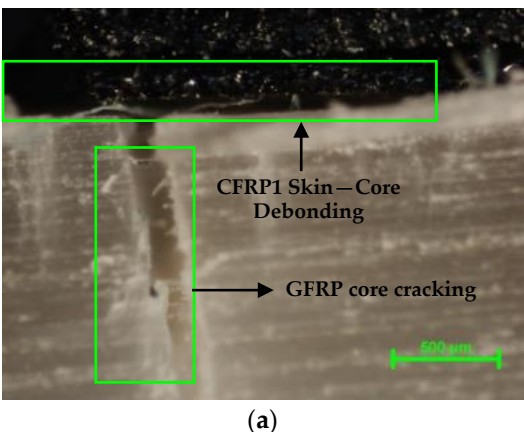 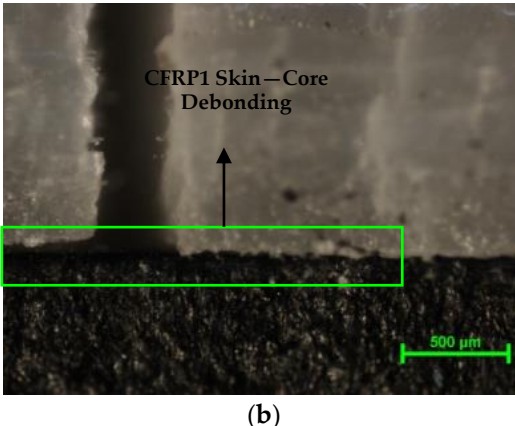

| (a) | (b) |

**Figure 14.** Failure modes of composite sandwich structure under bending testing (magnification 50×): (**a**) debonding between the upper skins and the core and complete cracking of the core; (**b**) debonding of the core from the lower skin.

Impact-tested sandwich specimens indicated complete breaking. The breaking of the composite sandwich specimens (Figure 15) was initiated from the level of the upper skin and propagated throughout the core structure, followed by complete breaking of the structure. The surface where the hammer hit the carbon skin came off the core, and the micrograph (Figure 15) represents the way of breaking on the opposite side of the hit where the carbon layer did not peel off the core.

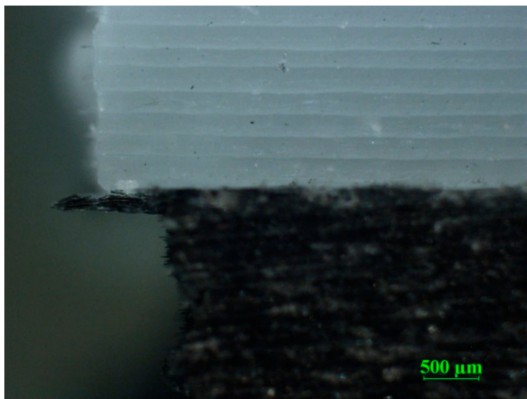

**Figure 15.** Failure modes of composite sandwich structure under impact testing (magnification 25×): breaking of the lower skin—the core of the sandwich structure.

The defects observed in Figure 16a (fiberglass core) are specific to parts manufactured by the FFF process and represent two of the most commonly encountered defects [12,25,50,51]: rectangular and triangular void formation. In Figure 16b, the main defects occurred in the carbon fiber filler (CFRP1), built with a filling density of 60%, were gaps and inter-track voids between the carbon fiber layers. Supplementary void formation (Figure 16b) may be created during the FFF process, as the filament is non-uniform, forming airgaps between tracks [12].

In the fiberglass core (Figure 17a) of the sandwich structure, we encounter the typical defects of the FFF process, but in a much smaller number compared to the carbon fiber skin (Figure 17b). In Figure 17a, the following types of defects were identified in the fiberglass core: inter-track voids between the deposition layers and gaps. Figure 17b shows inter-track voids between the fiberglass core and the carbon fiber skin. The void inter-track defect indicated a low degree of bonding between the two main components of the composite sandwich structures (core and skin). This defect explains a large part of the

failure modes, namely the debonding of the skin core, which appeared in the mechanical tests. In Figure 17b, the light white dots represent carbon fibers, and the light gray area is the material of the nylon matrix [52]. You can also see average hollow surfaces in extruded carbon filament layers. The interface between the carbon fiber layers was not clear, which indicated good adhesion between extruded filament layers.

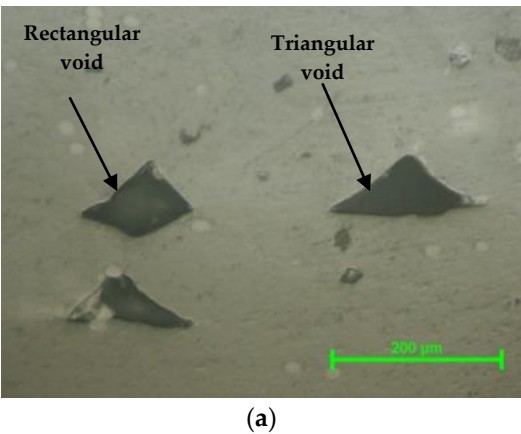

(**a**)

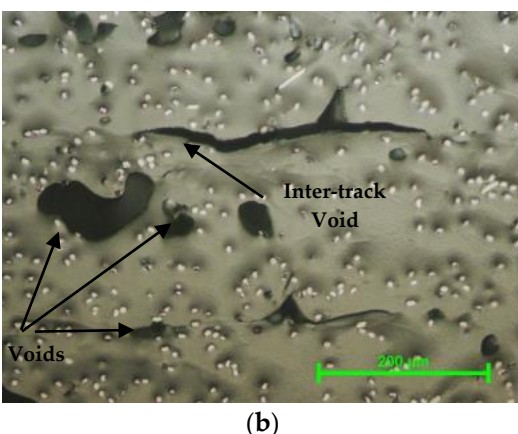

(**b**)

**Figure 16.** Analysis of defects in the cross section of the CFRP1-GFRP sandwich structure (magnification 200×): (**a**) defects occurring during the FFF process of the fiberglass core; (**b**) defects occurring during the FFF process of the carbon fiber skin.

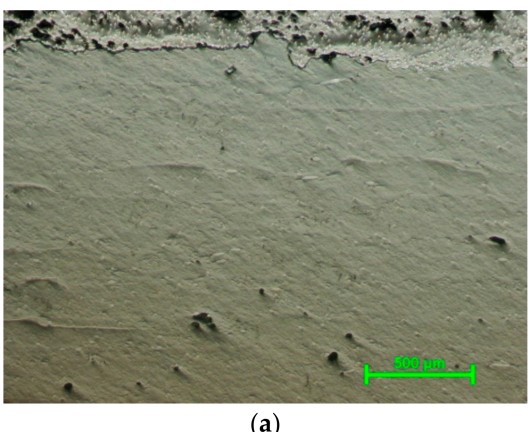

(**a**)

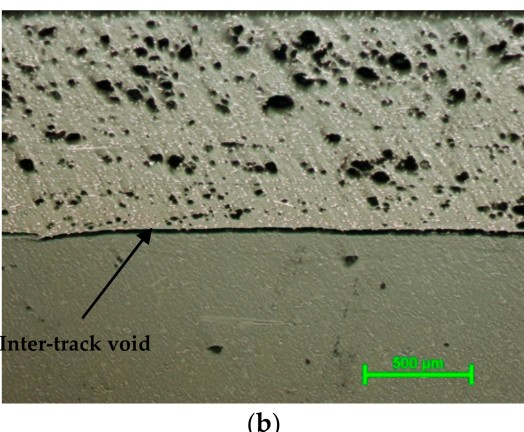

(**b**)

**Figure 17.** Analysis of defects in the cross section of the sandwich structure CFRP1-GFRP (magnification 50×): (**a**) defects occurring during the FFF process of the fiberglass core; (**b**) defects occurring during the FFF process at the carbon fiber skin and between the core and the skin.

Figure 18a shows the contour of the sandwich specimen, with a filling density of 20%, which showed minor defects between the layers of fiberglass, as well as the typical gaps between extruded glass filament layers. Figure 18b identifies layers with a 20% fill density and defects typical of the FFF process. Moreover, one can notice the poor adhesion between the successive layers on the height of extruded glass filament layers and between the two manufactured materials (CFRP1-GFRP).

The adhesion between the extruded layers is stronger if this area is heated above its sintering temperature. During the process of thermoplastic extrusion of the filament, the cooling is faster due to the short deposition times and, therefore, the adhesion between the 3D printed layers is weaker. To avoid this problem, deposition tracks have been developed to increase the heating time using air nozzles. This solution generated a high dimensional inaccuracy of the components manufactured by the FFF process. Some solutions for reducing the gaps in the FFF process of composite filaments are [53] infrared lamp heating,

hot air nozzle preheating, and plasma radiation. However, these solutions have led to large volume shrinkage [53].

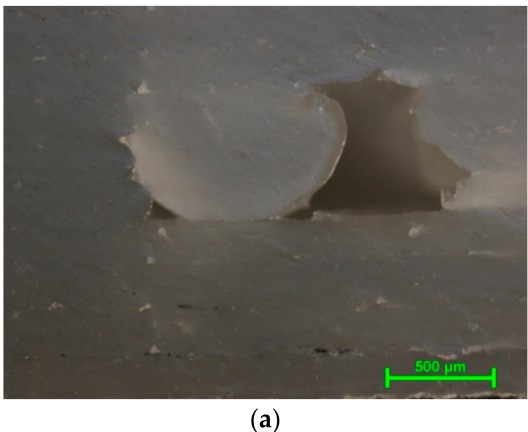
(**a**)

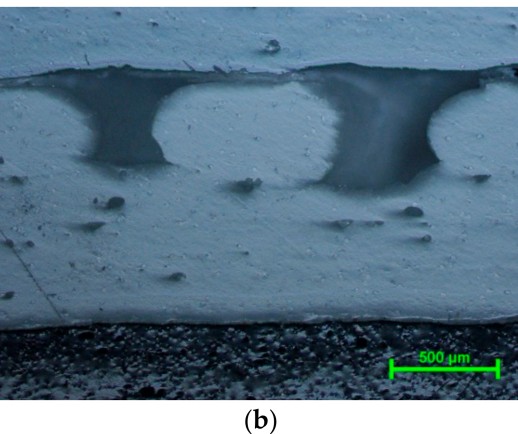
(**b**)

**Figure 18.** Analysis of defects in the cross section of the sandwich structure CFRP1-GFRP (magnification 50×): (**a**) defects occurring during the FFF process of the fiberglass core; (**b**) defects during the FFF process at the fiberglass core and at the CFRP1-GFRP interface.

### 3.7. Finite Element Analysis of the Fiber-Reinforced Composite Sandwich Structures

In the finite element analysis model, specimen size, material properties, punch radius and supports, and the distance between the supports was determined according to test on three-point bending and tensile loading. In this study, the reaction forces and equivalent stresses that occur when testing sandwich structures were investigated using the ANSYS 19.1 finite element analysis software system (ANSYS, Inc., Canonsburg, PA, USA). In the finite element analysis, the elastoplastic model was used for the components of composite sandwich structures, both for the two carbon fiber skins and for the fiberglass core.

The discretization of the models, for three-point bending and tensile strength, was performed with three-dimensional elements of Hexa type with an element size of 0.8 mm (Figure 19a,b). The punch and the two supports were discretized with the same type of Hexa element with a size of 2 mm (Figure 19a). For the components of the test machine (a punch and the two supports), rigid body properties were assigned in the analysis (which do not deform under the action of forces) [32]. Frictionless contacts have been established between the sandwich and punch skins and supports.

For the three-point bending tests, a displacement was applied in the middle of the composite sandwich specimen, which showed a value of 5 mm/min, under the same conditions as in the case of the experimental tests. Moreover, in the tensile tests of the composite sandwich specimens, a displacement of 5 mm/min was imposed in accordance with the experimental tests. The friction between the punch, supports, and the surface of the specimen was taken into account, and the coefficient of friction was 0.1 [54]. The two finite element models (three-point bending and tensile) were analyzed according to the properties of CFRP2-GFRP sandwich structures at a filling density of 100%.

For the finite element analysis of CFRP2-GFRP sandwich structures, several simplifying hypotheses were made [53–55]: the constituents have a linear elastic behavior, the matrix (polyamide) has isotropic properties; chopped fibers are transverse isotropic; the fiber matrix has a perfect adhesion and does not show gaps or defects present in the tested sandwich specimens. The modulus of elasticity, used in the FEA model, for CFRP1 has the value of 8110 MPa, and for the fiberglass core the value is 4000 MPa. The Poisson's ratio was assumed to be that of the composite filament, so that for the carbon fiber skin, it had the value of 0.3 [55,56], and for the fiberglass, 0.25 [57].

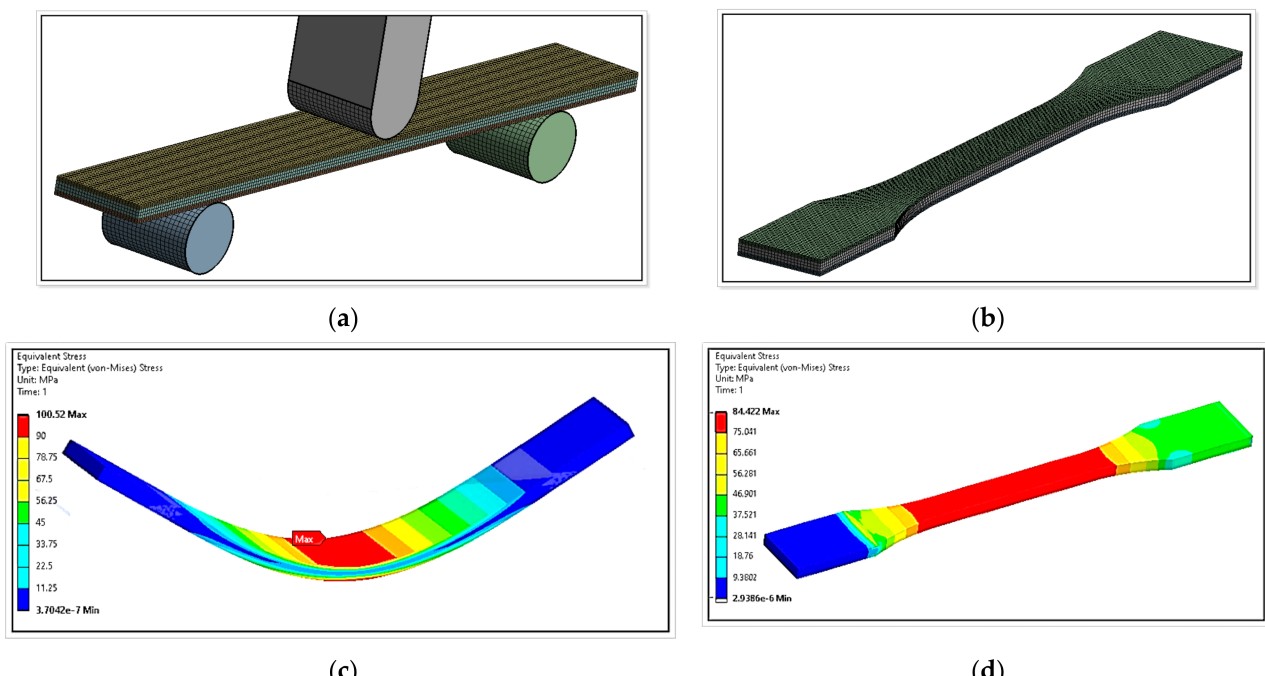

**Figure 19.** Finite element analysis: (**a**) FEA model—three-point bending test; (**b**) FEA model—tensile test; (**c**) equivalent stress distribution of composite sandwich subjected to three-point bending load; (**d**) equivalent stress distribution of the composite sandwich subjected to tensile load.

Finite element analysis of composite sandwich structures was used following two aspects: comparative study between the failure behavior of composite sandwich specimens, tested at three-point bending/tensile, and the result obtained from finite element analysis of the same specimens; comparative study of the maximum forces appearing at the rupture of the specimens tested at bending in three points/tensile and the reaction forces, appearing in the structure of the supports, from the model with finite elements. Thus, following the investigation of the specimens subjected to experimental tests and finite element analysis, it can be stated that the rupture occurs, in both cases, in the middle of the composite sandwich specimen, where the tension is maximum (Figure 19c,d). The equivalent von Mises stress, at the request of bending in three points, has the maximum value (100.5 MPa), in the middle area of the specimen, and is close to the mean value of bending strength (112.6 MPa) of the sandwich specimen obtained from experimental tests. Moreover, the value of the equivalent stress at tensile loading (84.4 MPa) showed a value close to the experimentally obtained mean tensile strength (75 MPa).

The result of the comparative study (Table 11), between the maximum forces, appeared in the experimental tests (three-point bending and tensile) and the reaction forces, appeared in the supports of the finite element model and represented adequate validation. The errors that occurred between experimental and simulated results of the finite element method were of maximum 5.2%. These errors may be due to the simplifying assumptions used in the finite element analysis of 3D printed sandwich structures.

**Table 11.** Comparative analysis between experimental results and results obtained from FEA.

| Test Type | Reaction Forces-Experimental [kN] | Reaction Forces-FEA [kN] | Relative Error (%) |
|---|---|---|---|
| Three-point bending | 0.19 | 0.20 | 5.20 |
| Tensile | 2.50 | 2.62 | 4.80 |

## 4. Conclusions

Composite sandwich structures, manufactured by the material extrusion 3D printing processes, can be successfully used as components (wing leading edge, fuselage structure) of unmanned aircraft or drones due to their high strength-to-mass ratio and high flexural rigidity and bending strength yet low mass. This paper presents a study on the mechanical properties of sandwich structures composed of carbon fiber and fiberglass, manufactured by the FFF process, subjected to mechanical tests (three-point bending tests, tensile strength tests and impact tests). For these composite sandwich structures, the mechanical performances were determined using different infill densities (100%, 60% and 20%), two types of carbon fibers for the skins, and the same core material (glass fiber).

The results of the mechanical tests, the sandwich structure CFRP2-GFRP, has superior performance, compared to the sandwich structure CFRP1-GFRP, for all the infill densities analyzed. The high performance of the CFRP2-GFRP sandwich structure is due to the superior performance of the base material used in the skins. Thus, for the CFRP1-GFRP sandwich structure, co-polymer PA6/66 was used as base material (matrix), and for the CFRP2-GFRP sandwich structure, polyamide (nylon) was used. The polyamide (nylon) filament has 15% higher values for mechanical properties compared to the PA6/66 copolymer. In the case of three-point bending and impact tests, the strength is directly proportional to the infill density, whereas in tensile strength tests the strength values do not change significantly with the variation of the infill density.

From microscopic analysis of the breaking surfaces, the typical defects of 3D printed composite filaments were identified. The microstructure of the cross sections of the sandwich specimens manufactured by the FFF process was examined using an optical microscope, where gaps and inter-track voids between the layers of extruded material were identified. These hollow contents and inter-track voids are the main cause of the lower mechanical properties of 3D printed sandwich structures compared to conventionally manufactured carbon/glass-fiber-reinforced composite sandwich structures. Moreover, the experimental results of the three-point bending and tensile tests of the 3D printed sandwich specimens were validated, making a comparison between the reaction forces that appeared in the experimental tests and the reaction forces of the FEA model, resulting in maximum errors of 5.2%.

This study shows that the mechanical properties of composite sandwich structures manufactured through the FFF process, with three filling densities (100%, 60%, and 20%), are sufficient for the implementation of these types of structures in various prototypes from areas such as aerospace and automotive.

**Author Contributions:** Conceptualization, S.-M.Z., M.A.P. and G.R.B.; methodology, M.A.P., L.-A.C. and I.S.P.; software, C.L.; validation, V.-M.S., investigation, S.-M.Z., M.A.P. and G.R.B.; writing—original draft preparation, G.R.B., S.-M.Z., L.-A.C. and I.S.P.; project administration, S.-M.Z. All authors have read and agreed to the published version of the manuscript.

**Funding:** This work was supported by a grant of the Romanian Ministry of Education and Research, CCCDI-UEFISCDI, project number PN-III-P2-2.1-PED-2019-0739, within PNCDI III. We also acknowledge to PRO-DD Structural Founds Project (POS-CCE, O.2.2.1., ID 123, SMIS 2637, ctr. no 11/2009) for providing the infrastructure used in this work.

**Conflicts of Interest:** The authors declare no conflict of interest.

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
