# Peer review of "Fabrication and Characterization of Fiber-Reinforced Composite Sandwich Structures Obtained by Fused Filament Fabrication Process"

_coatings, doi:10.3390/coatings11050601_

Round 1

Reviewer 1 Report

Dear Sirs,

The article entitled “Fabrication and Characterization of Fiber-Reinforced Composite Sandwich Structures Obtained by Fused Filament Fabrication Process” presents a study on the mechanical properties of sandwich structures composed of carbon fiber and fiberglass, manufactured by the FFF process, subjected to mechanical tests (three-point bending tests, tensile strength tests and impact tests). Content seems to be relatively new and promising. But please reinforce your novelty and comment the work from doi: 10.1016/j.matdes.2017.10.021, and include that in your manuscript. The study is well-written, but not so directed at a prospective application at times. That should be corrected. Minor grammatical mistakes are also present throughout the text. Please revise them. Other than that, I add some small remarks:

Abstract:

“The application of fused filament fabrication processes is rapidly expanding in many 11 domains, mainly due to the flexibility of manufacturing structures with complex geometries in a 12 short time” – missing examples of applications

A final sentence summarizing the overall achievements of this article, novelty and applicability are missing.

Introduction:

Line 88: throw -> through?

Why CFRP and GFRP? What is your goal in concrete? What are you aiming at?

Results and Conclusions:

“used for components (wing leading edge, fuselage 500 structure) of unmanned aircraft or drones” – so, in what concerns the defects observed, could your structures be still accepted for real-life use, namely as the aforementioned components? Were the mechanical tests adapted to such circumstances?

Again, you should finish your manuscript in a different manner, summarizing what you have accomplished and reinforcing your contribution to scientific knowledge, and to society in general, namely those that could directly benefit from your research.

Reviewer 2 Report

  1. Include a statistics section and mention the sample number used for each test and the statistic method used to evaluate those results for significance of difference.
  2. Figure 9, 10, 11- error bars are not included. 
  3. Remodify the figures for consistency in presentation with type and color for groups.
  4. Elaborate the figure labels. Provide atleast two sentences describing the figure. Also mention the sample number used and statistic method used in each figure label.
  5. Discuss the crack propagation and the steps needed to avoid the gaps and defects observed in this study.
  6. Provide a cartoon depicting the printer and printing process for easy understanding for readers.

Reviewer 3 Report

The manuscript ‘Fabrication and Characterization of Fiber-Reinforced Composite Sandwich Structures Obtained by Fused Filament Fabrication Process’ presents the characterization of fiber-reinforced composites for two skin materials and three infill densities by fused filament fabrication.  The results show the potential to create composites with low mass and high mechanical strength. The manuscript can be published with minor revisions.

1. The manuscript mentioned the skin material 2 has ‘15% higher mechanical properties’ compared to skin material 1. Can the authors show how the quantitive comparison is obtained from Tables 1-2.

2. Can the authors explain the reason for the insignificant change of the tensile strength values with the variation of the infill density, and also the irregularity of the displacement between samples CFRP-1-GFRP-20% and 60% in Figure 8.

3. Defect discussions. What are the defects in Figure 16(a)?

The defects are identified in the manuscript. It would be better for the paper if solutions/optimizations to the defects are offered.

4.Minor errors:

Line 121-122: ‘used in the first type of sandwich structures that was built on flat XX direction, in Table 1are shown’. Need to be rephrased.

Line 486: b))
